# Radiological Variability in Pancreatic Neuroendocrine Neoplasms: A 10-Year Single-Center Study on Atypical Presentations and Diagnostic Challenges

**DOI:** 10.3390/biomedicines13020496

**Published:** 2025-02-17

**Authors:** Eleanor Danek, Helen Kavnoudias, Catriona McLean, Jan F. Gerstenmaier, Bruno Di Muzio

**Affiliations:** 1Department of Radiology, The Alfred Hospital, Melbourne, VIC 3004, Australia; 2Department of Anatomical Pathology, The Alfred Hospital, Melbourne, VIC 3004, Australia

**Keywords:** pancreatic neuroendocrine neoplasm (PNEN), pancreatic neuroendocrine tumor (PNET), hyperenhancing pancreatic lesions, cystic pancreatic lesions

## Abstract

**Background:** Pancreatic neuroendocrine neoplasms (PNENs) are rare but clinically significant tumors with variable radiological presentations that complicate diagnosis. While typical PNENs are well characterized, atypical features, such as cystic or hypoenhancing patterns, are less understood and can lead to diagnostic delays or misdiagnosis. This study aimed to evaluate atypical radiological presentations of PNENs, focusing on their impact on diagnostic pathways and differentiation from other pancreatic pathologies. **Methods:** A retrospective review was conducted of all PNEN cases diagnosed at a single tertiary center between 2010 and 2020. Cases with histopathological confirmation and available cross-sectional imaging were included. Radiological features were categorized as typical (solid and hyperenhancing) or atypical (cystic and hypoenhancing). Demographic, radiological, and pathological data were analyzed. Comparisons between typical and atypical PNENs were performed using descriptive and inferential statistics. **Results:** Among 77 PNEN cases, 39 met the inclusion criteria. Atypical radiological presentations were identified in 46% of cases, including cystic (18%) and hypoenhancing (28%) lesions. Hypoenhancing PNENs were significantly more likely to present with advanced disease (54% vs. 14% in typical PNENs, *p* = 0.016). In contrast, none of the cystic PNENs exhibited advanced disease. Atypical PNENs posed greater diagnostic challenges, with alternative diagnoses initially considered in 64% of hypoenhancing and 43% of cystic cases compared to 10% of typical PNENs (*p* = 0.0042). **Conclusions:** Atypical PNENs, particularly hypoenhancing lesions, present significant diagnostic challenges and are more likely to be associated with advanced disease. These findings highlight the need for improved recognition of atypical imaging patterns and more precise diagnostic strategies. However, the retrospective design and small cohort size limit the generalizability of our findings. Further multicenter studies are warranted to refine the imaging criteria and optimize the differentiation from other pancreatic neoplasms.

## 1. Introduction

Pancreatic neuroendocrine neoplasms (PNENs) are rare tumors originating from the embryonic endodermal ductal–acinar system, accounting for 1–2% of all clinically detected pancreatic neoplasms, with an incidence of 2 to 6 per 100,000 individuals [1,2,3,4]. They exhibit a slight male predominance and are most commonly diagnosed in the sixth decade of life [5,6]. While the majority of PNENs occur sporadically, approximately 10% are associated with hereditary syndromes that are predisposed to neuroendocrine tumor development [7]. Prognosis is highly variable, with five-year survival rates exceeding 90% for localized, well-differentiated tumors but significantly worse in cases of distant metastases or poorly differentiated histology [2,3,4,5,6].

Advances in cross-sectional imaging have led to a more than tenfold increase in PNEN detection, particularly in asymptomatic, non-functional cases [3,4,5]. Nevertheless, diagnosing PNENs can be challenging due to their variable imaging characteristics.

Most PNENs are traditionally characterized as well-defined, hypervascular, solid masses, allowing for a confident diagnosis in the majority of cases based on these “typical” features [6,8]. However, a subset of tumors deviates from this classical pattern, presenting as cystic or hypoenhancing lesions [8,9]; these “atypical” features often overlap with other pancreatic neoplasms, such as pancreatic ductal adenocarcinoma or pancreatic cystic neoplasms [7,8,10], complicating radiological differentiation. As a result, atypical PNENs may be underrecognized or misdiagnosed, leading to delays in definitive diagnosis and treatment. Despite their clinical significance, limited data exist on atypical imaging presentations, underscoring the need for further investigation.

This study aims to address these gaps by evaluating the atypical radiological presentations of PNENs, particularly cystic and hypoenhancing lesions, and assessing their impact on differential diagnosis and diagnostic pathways. By correlating histopathological findings with imaging features, we aim to provide a deeper understanding of the relationship between tumor morphology and its atypical presentations. Additionally, by comparing the demographic, radiological, and pathological characteristics of atypical PNENs with their typical counterparts, we seek to improve recognition of these patterns, support earlier and more accurate diagnoses, and refine diagnostic strategies to ultimately enhance patient outcomes.

## 2. Materials and Methods

This retrospective, single-center study was conducted at a tertiary care hospital with institutional ethics committee approval (Project No: 774/19). A search of the Radiology Information System (RIS) was performed from January 2010 to January 2020 to identify cases diagnosed with pancreatic neuroendocrine neoplasms (PNENs). Cases were identified using diagnostic codes and relevant keywords (e.g., “pancreatic neuroendocrine tumor”, “PNET”, and “PNEN”) to screen imaging reports suggestive of PNENs.

The inclusion criteria were (1) availability of cross-sectional imaging (CT or MRI) for review and (2) histopathological confirmation of PNEN via biopsy or surgical resection. Cases lacking imaging or histopathological data were excluded.

Diagnostic CT scans were reviewed by a senior radiologist with over 10 years of experience in abdominal imaging. Imaging features were categorized as typical (solid and hyperenhancing) or atypical (cystic and hypoenhancing) based on standardized criteria. Cystic PNENs were defined as lesions with ≥50% of their volume demonstrating fluid-like low attenuation (<20 HU), while hypoenhancing PNENs were defined as those with less enhancement than the surrounding pancreas during the arterial and portal venous phases.

Advanced disease was defined as either the presence of distant metastatic disease identified on cross-sectional imaging (CT/MRI) or PET-CT studies or locally advanced (LA) tumors deemed unresectable based on established pancreatic resectability criteria.

The pathology specimens were evaluated by a senior pathologist according to the 2017 WHO classification, with the Ki-67 proliferation index and mitotic counts assessed where available.

Descriptive statistics were used to summarize the demographic, radiological, and pathological characteristics. Categorical variables were presented as frequencies and percentages, while continuous variables were summarized as medians with interquartile ranges (IQR). Group comparisons between typical and atypical PNENs were conducted using Fisher’s exact test for categorical variables and the Mann–Whitney *U* test for non-normally distributed continuous variables. For comparisons of continuous variables across multiple PNEN subtypes, one-way analysis of variance (ANOVA) was used when the normality assumption was met. A *p*-value < 0.05 was considered statistically significant.

The study adhered to institutional ethical guidelines and the Declaration of Helsinki, and all patient data were anonymized prior to analysis.

## 3. Results

### 3.1. Case Detection and Demographics

A total of 77 PNEN cases were identified over the study period, of which 39 met the inclusion criteria (Table 1). Cross-sectional imaging was available for all included cases, with CT being the initial modality in 90% and MRI in 10%. Additional imaging with ^68^Ga-DOTATATE PET or FDG-PET was performed in 82% of cases. Typical PNENs accounted for 54% of the cohort and were characterized by well-defined, hypervascular solid lesions (Figure 1, Figure 2 and Figure 3). In contrast, atypical PNENs comprised 46% (*n* = 18), including 7 cystic lesions (18%) and 11 hypoenhancing lesions (28%).

Among the 39 cases, 69% were male, with a median age of 61 years (IQR: 53–66). Gender distribution did not significantly differ between PNEN subtypes, as atypical PNENs showed a higher proportion of male cases (77.8%) compared to typical PNENs (61.9%), but this difference was not statistically significant (Fisher’s exact test, *p* = 0.322). Male predominance was observed in both typical and atypical PNENs. The age distribution varied across subtypes, with cystic PNENs presenting in a younger demographic (median age 56 years, IQR: 46–66) compared to hypoenhancing PNENs (65 years, IQR: 60–86) and typical PNENs (66 years, IQR: 56–75), although these differences were not statistically significant (ANOVA, *p* = 0.550).

Most cases (51%) were incidentally detected, while 44% presented symptoms such as abdominal pain, obstructive cholangitis, or hypoglycemia.

### 3.2. Radiological Characteristics

Atypical radiological characteristics were observed in 46% of cases, with 18% cystic and 28% hypoenhancing. In contrast, 54% displayed a typical solid, hyperenhancing CT or MRI presentation. The median tumor size was 2.4 cm (IQR: 1.6–4.0 cm), with atypical PNENs being larger than typical PNENs (Table 2).

Tumor sizes varied significantly among the PNEN subtypes (ANOVA, *p* = 0.025). Hypoenhancing PNENs had significantly larger tumors compared to typical PNENs (Tukey’s post hoc test, *p* = 0.019). However, no significant differences were observed between the cystic PNENs and the other subtypes.

Tumor distribution was predominantly in the pancreatic tail (41%), followed by the body (33%) and the head/uncinate process (26%). Pancreatic duct dilatation and parenchymal atrophy were identified in 10% of cases, while tumor calcifications were present in 15%.

Advanced disease, defined as either locally advanced disease according to the resectability criteria or the presence of metastatic disease, was observed in 21% of the cases overall. The prevalence of advanced disease varied significantly across the PNEN subtypes. Among the typical PNENs, 14% were classified as having advanced disease. In contrast, hypoenhancing PNENs demonstrated a markedly higher prevalence, with 54% of the cases presenting with either locally advanced or metastatic disease. Notably, no cases of advanced disease were observed in the cystic PNENs.

Symptomatic presentation was more frequent in the cystic PNENs (57% vs. 29% in the typical PNENs) (Figure 4 and Figure 5), and the lesions were larger (median size 2.5 cm vs. 1.7 cm). Calcifications and duct dilatation were also more common in the cystic PNENs (29% vs. 15% for calcifications and 14% vs. 10% for duct dilatation). Importantly, none of the cystic PNEN cases demonstrated locoregional invasion or metastatic disease.

Most cases of hypoenhancing PNENs (*n* = 11) presented symptomatically (64%), and the lesions were significantly larger (median size 4.8 cm vs. 1.7 cm for typical PNENs). Calcifications were observed in 9% of hypoenhancing PNENs. As pointed out, these tumors were significantly more likely to present with advanced disease compared to other PNEN subtypes (Fisher’s exact test, *p* = 0.016) (Figure 6 and Figure 7).

### 3.3. Diagnostic Challenges and Alternative Diagnoses

Atypical PNENs were associated with higher diagnostic uncertainty compared to typical lesions. Initial imaging suggested alternative diagnoses in 64% of hypoenhancing PNENs and 43% of cystic PNENs compared to only 10% of typical PNENs. Fisher’s exact test confirmed a statistically significant association between PNEN type (typical vs. atypical) and diagnostic uncertainty (*p* = 0.0042). Hypoenhancing PNENs were commonly mistaken for pancreatic adenocarcinoma, while cystic PNENs were frequently confused with intraductal papillary mucinous neoplasms (IPMNs) or mucinous cystic neoplasms. Further dedicated pancreas protocol imaging improved lesion characterization in two-thirds of the cases.

### 3.4. Pathological Characteristics

Pathology specimens were available for 20 cases (51%) and classified according to the 2017 World Health Organization (WHO) criteria for pancreatic neuroendocrine neoplasms [11,12]. Among the typical PNENs, 67% were Grade 1 and 33% were Grade 2 (Figure 8 and Figure 9). Notably, no cases of Grade 3 or poorly differentiated neoplasms were identified in our small cohort.

In the cystic PNENs, 83% were Grade 1 and 17% were Grade 2. In contrast, the hypoenhancing PNENs demonstrated a more aggressive profile, with 60% classified as Grade 2 and 40% as poorly differentiated pancreatic neuroendocrine carcinomas (PNECs) (Table 3).

The median Ki-67 index was 1% for the typical and cystic PNENs. However, the hypoenhancing PNENs demonstrated a significantly higher Ki-67 index, with a median value of 20%, reflecting their more aggressive biological behavior.

While the analysis revealed that the hypoenhancing PNENs demonstrated a higher proportion of aggressive pathological features (60% classified as Grade 2 or poorly differentiated), the small sample size (*n* = 20) limits the statistical power to detect significant differences. The trends observed, such as the higher median Ki-67 index in hypoenhancing tumors (20% vs. 1% for typical and cystic PNENs), suggest a potential association warranting further investigation in larger cohorts.

## 4. Discussion

### 4.1. Overview of the Key Findings and Interpretations

This study is among the few to stratify pancreatic neuroendocrine neoplasms (PNENs) based on radiological characteristics, comparing “typical” hypervascular solid lesions (Figure 3) to “atypical” cystic or hypoenhancing PNENs. Our findings challenge the conventional perception that PNENs are predominantly hypervascular, showing that atypical presentations accounted for nearly half (46%) of cases.

Atypical PNENs posed significant diagnostic challenges, with initial imaging misclassification rates of 64% for hypoenhancing PNENs and 43% for cystic PNENs compared to only 10% for hypervascular PNENs. This statistically significant association highlights the difficulty in distinguishing atypical PNENs from other pancreatic neoplasms, which is consistent with prior studies that have highlighted the diagnostic overlap [8,9]. Hypoenhancing PNENs were most frequently misdiagnosed as pancreatic adenocarcinoma, while cystic PNENs were often confused with intraductal papillary mucinous neoplasms (IPMNs) or mucinous cystic neoplasms. Importantly, dedicated pancreas protocol imaging led to improved lesion characterization in two-thirds of cases, underscoring the need for specialized imaging techniques to refine differential diagnoses.

Our findings align with prior studies reporting the underappreciated prevalence of atypical PNENs, particularly cystic lesions [8,13,14,15,16,17,18,19,20]. While some studies suggest no size differences between cystic and solid PNENs [8,14,18], our results align with reports showing that cystic PNENs are typically larger than their solid counterparts at presentation [13,15,16,20].

Importantly, none of the cystic PNENs in our cohort demonstrated locally advanced or metastatic disease, differing from studies indicating no differences between cystic and solid PNENs [16,18]. Notably, most cystic PNENs follow a benign, indolent course, comparable to their typical solid counterparts, particularly when smaller than 2 cm [21,22].

Hypoenhancing PNENs presented with more aggressive features, including larger size, locally advanced disease, and metastases, consistent with the literature linking hypoenhancing lesions to poorly differentiated neoplasms or neuroendocrine carcinomas [9,10,23]. The male preponderance and association with advanced grading highlight the importance of considering PNENs as a differential diagnosis for hypoenhancing pancreatic lesions, particularly in older men where adenocarcinoma is suspected.

### 4.2. Clinical Implications

Recognizing atypical PNEN presentations is crucial for improving early diagnosis and management. Hypoenhancing PNENs, in particular, require prompt recognition due to their aggressive behavior and metastatic potential, while cystic PNENs, though less aggressive, are prone to misdiagnosis, potentially leading to unnecessary interventions or treatment delays. The presence of a hypervascular rim, even in predominantly cystic lesions, should prompt further evaluation with MRI or functional imaging.

These diagnostic challenges underscore the need for heightened clinical suspicion and optimized diagnostic pathways. Advanced imaging modalities, such as ⁶⁸Ga-DOTATATE PET/CT and endoscopic ultrasound (EUS), are essential for further workup distinguishing atypical PNENs from other pancreatic neoplasms and guiding appropriate management [7,24,25].

### 4.3. Study Limitations and Future Directions

This study has several limitations. First, its retrospective, single-center design may limit generalizability. The modest sample size, reflective of the rarity of pancreatic neuroendocrine neoplasms, may have impacted the statistical power of the subgroup analyses and limited the ability to draw definitive conclusions about less common imaging features. The absence of standardized imaging protocols for earlier cases and incomplete pathological grading data may have influenced the accuracy of radiological–pathological correlations. Given the relatively small cohort, some imaging patterns observed may not fully represent the spectrum of PNEN presentations seen in broader clinical practice.

Second, there is an inherent selection bias associated with conducting this study at a tertiary referral center. This setting likely resulted in an overrepresentation of diagnostically complex, atypical, or advanced cases. Consequently, the prevalence of atypical PNEN subtypes in this study may be higher than in general clinical populations.

Furthermore, while biopsy-based diagnoses were included, we acknowledge that they may not capture all histopathological features, such as lymphovascular invasion or nodal metastases. However, given the study’s limited scope, this did not significantly impact our primary research objectives.

Future research should focus on multicenter studies with larger cohorts to validate the prevalence and clinical significance of atypical PNEN presentations. Longitudinal studies assessing the correlation of imaging features with clinical outcomes, including survival and treatment response, would provide valuable insights. The role of advanced imaging techniques, such as radiomics and artificial intelligence, in differentiating PNEN subtypes warrants further exploration.

## 5. Conclusions

Our study highlights the variability in PNEN radiological presentations, with atypical cystic and hypoenhancing lesions comprising a substantial proportion of cases. These atypical forms pose diagnostic challenges due to their overlap with other pancreatic neoplasms, particularly hypoenhancing PNENs, which are associated with aggressive behavior and advanced disease.

Although limited by its retrospective design, small cohort size, and tertiary referral bias, this study provides valuable insights into atypical PNEN imaging patterns. Greater clinical awareness of these variants may help reduce misdiagnosis and unnecessary interventions, ultimately improving patient care. Further multicenter studies are needed to refine the imaging criteria and diagnostic strategies.

## Figures and Tables

**Figure 1 biomedicines-13-00496-f001:**
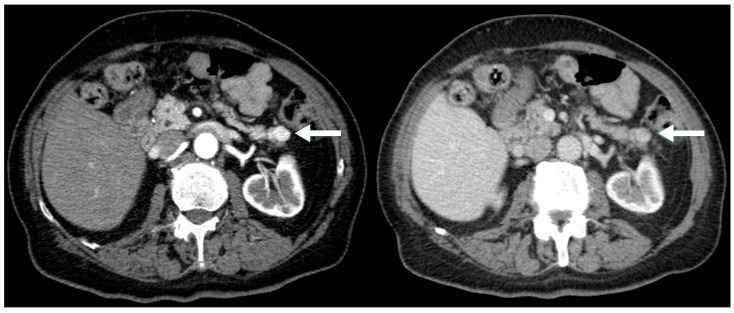
A 77-year-old male with an incidental pancreatic mass detected (arrow) on a trauma CT scan. Contrast-enhanced CT images show a 1.7 cm arterially hyperdense lesion in the tail of the pancreas (**left**, arterial phase). The lesion remains hyperdense relative to the background pancreas on the portal venous phase (**right**).

**Figure 2 biomedicines-13-00496-f002:**
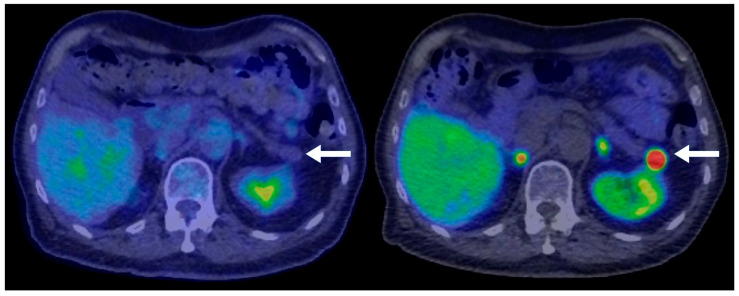
A 77-year-old male with an incidental pancreatic mass detected on a trauma CT scan. PET-CT demonstrates that the lesion (arrows) is non-FDG-avid (**left**) but intensely ^68^Ga-DOTATATE-avid (**right**), consistent with a well-differentiated neuroendocrine tumor. No evidence of metastatic disease on the PET-CT.

**Figure 3 biomedicines-13-00496-f003:**
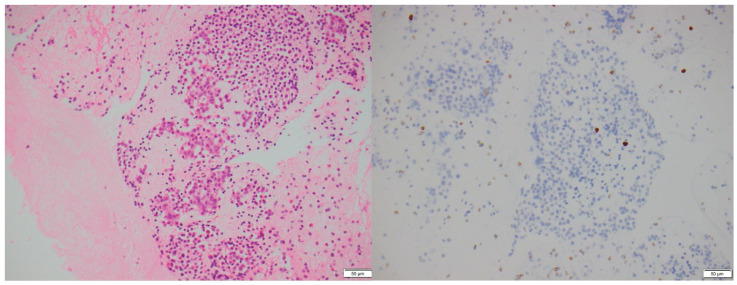
A 77-year-old male with an incidental pancreatic mass detected on a trauma CT scan. H&E stain (**left**): Fine-needle aspirate of the pancreas showing a Grade 1 PNEN with a nested architecture, minimal nuclear atypia, and moderate cytoplasm. Ki-67 immunohistochemistry (**right**): Ki-67 index < 2%, confirming a low proliferative rate (×200 magnification).

**Figure 4 biomedicines-13-00496-f004:**
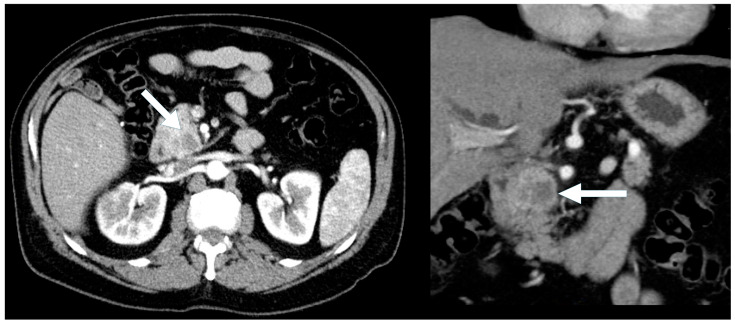
A 56-year-old man presenting with painless jaundice. Contrast-enhanced CT demonstrates a pancreatic head mass (arrow) with mixed solid and cystic/necrotic components, showing arterial phase hyperenhancement. (left) Axial arterial phase CT. (right) Coronal arterial phase CT.

**Figure 5 biomedicines-13-00496-f005:**
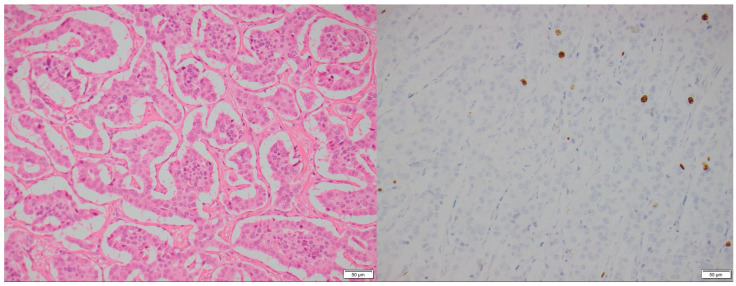
A 56-year-old man presenting with painless jaundice. H&E stain (**left**): Biopsy of the pancreatic mass demonstrating a Grade 1 PNEN with a nested architecture, minimal nuclear atypia, and moderate cytoplasm. Ki-67 immunohistochemistry (**right**): Ki-67 index < 1%, indicating a low proliferative rate (×200 magnification).

**Figure 6 biomedicines-13-00496-f006:**
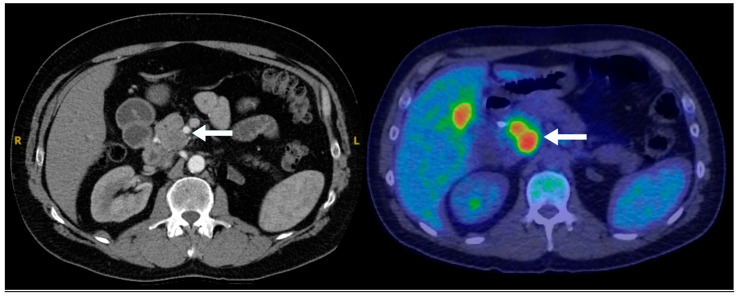
A 47-year-old male presenting with biliary obstruction. CT (**left**): Arterial phase imaging shows a 2.5 cm hypodense mass in the pancreatic head (arrow). PET-CT (**right**): Performed after ERCP and biliary stent placement, demonstrating an intensely FDG-avid pancreatic head mass (arrow), consistent with a poorly differentiated primary neuroendocrine tumor. Metastatic disease involving porta hepatis and peripancreatic lymph nodes, as well as liver metastases, is also evident on the PET.

**Figure 7 biomedicines-13-00496-f007:**
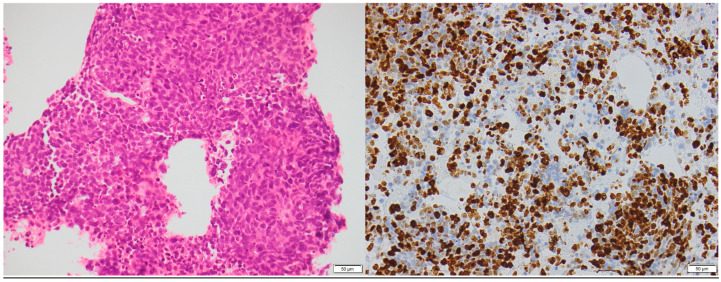
A 47-year-old male presenting with biliary obstruction. H&E stain (**left**): Biopsy of the pancreatic mass demonstrating a poorly differentiated pancreatic neuroendocrine carcinoma (PNEC) with high cellularity, hyperchromatic nuclei, high nuclear-to-cytoplasmic ratio, and apoptosis. Ki-67 immunohistochemistry (**right**): A high Ki-67 index, consistent with a high-grade neuroendocrine carcinoma (×200 magnification).

**Figure 8 biomedicines-13-00496-f008:**
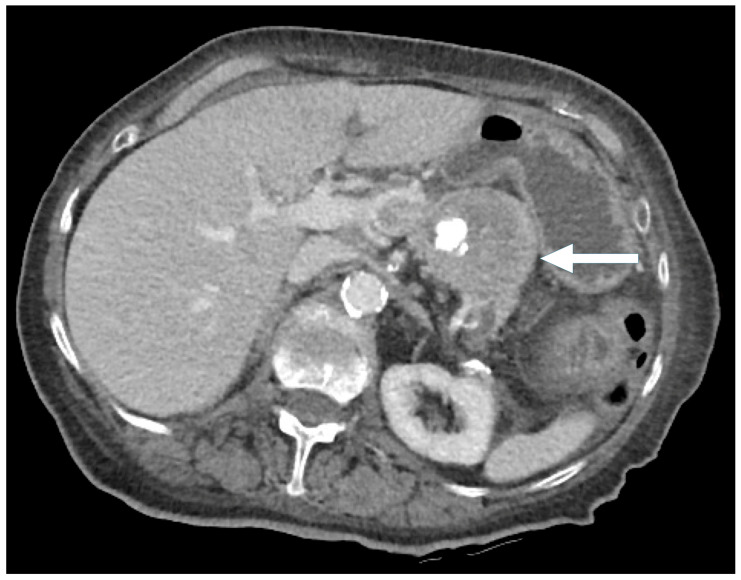
An 86-year-old female presenting with hypercalcemia and fatigue. CT: Pancreatic tail mass with mixed solid and cystic/necrotic components (arrow), along with focal gross calcification.

**Figure 9 biomedicines-13-00496-f009:**
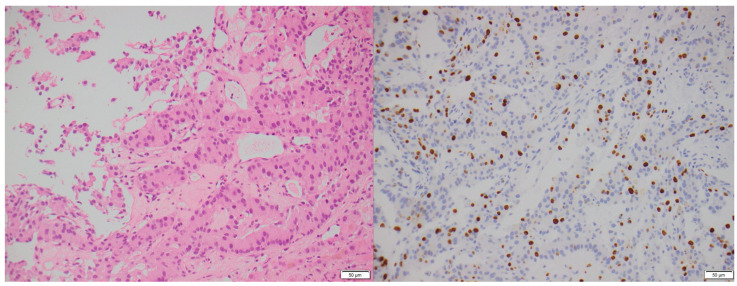
An 86-year-old female presenting with hypercalcemia and fatigue. H&E stain (**left**): Liver core biopsy showing a Grade 2 pancreatic neuroendocrine tumor (PNEN) with nested architecture, minor nuclear variation, and moderate cytoplasm. Ki-67 immunohistochemistry (**right**): a 20% Ki-67 index, consistent with intermediate proliferative activity (×200 magnification).

**Table 1 biomedicines-13-00496-t001:** Patient demographics in typical vs. atypical pancreatic neuroendocrine neoplasms.

	All Cases (*n* = 39)		Typical (*n* = 21)		Atypical (*n* = 18)			
					Cystic		Hypoenhancing	
	Number	Percent	Number	Percent *	Number	Percent *	Number	Percent *
Total	39	100	21	54	7	18	11	28
Age (years, median)	65	-	66	-	56	-	65	-
Sex								
Male	27	69	13	62	5	71	9	82
Female	12	31	8	38	2	29	2	18
Symptoms at diagnosis	17	44	6	29	4	57	7	64
Incidental diagnosis	20	51	13	62	3	43	4	36
Genetic predisposition	2	5	2	10	0	0	0	0

* Percentages are calculated using the respective subcategory (e.g., ‘Cystic’ or ‘Hypoenhancing’) as the denominator.

**Table 2 biomedicines-13-00496-t002:** Radiological characteristics of typical vs. atypical pancreatic neuroendocrine neoplasms.

	All Cases (*n* = 39)		Classical (*n* = 21)		Atypical (*n* = 18)			
					Cystic		Hypoenhancing	
	Number	Percent	Number	Percent *	Number	Percent *	Number	Percent *
Total	39	100	21	54	7	18	11	28
Size (cm, median)	2.4	-	1.7	-	2.5	-	4.8	-
Location								
Head/uncinate process	10	26	4	19	2	29	4	36
Body/neck	13	33	10	48	1	14	2	18
Tail	16	41	7	33	4	57	5	45
Calcification	6	15	3	14	2	29	1	9
Duct dilatation	4	10	2	10	1	14	1	9
Distal atrophy	4	10	3	14	0	0	1	9
LA or metastatic disease	9	23	3	14	0	0	6	54
Alternative diagnosis	12	31	2	10	3	43	7	64

* Percentages are calculated using the respective subcategory (e.g., ‘Cystic’ or ‘Hypoenhancing’) as the denominator.

**Table 3 biomedicines-13-00496-t003:** Pathological grading in the typical vs. atypical pancreatic neuroendocrine neoplasms.

	All Cases (*n* = 39)		Classical (*n* = 21)		Atypical (*n* = 18)			
					Cystic		Hypoenhancing	
	Number	Percent *	Number	Percent *	Number	Percent *	Number	Percent *
Pathology report	20	51	9	43	6	87	5	45
WHO grade								
Grade 1 NET	11	55	6	67	5	83	0	0
Grade 2 NET	7	35	3	33	1	17	3	60
Grade 3 NET	0	0	0	0	0	0	0	0
NEC	2	10	0	0	0	0	2	40
Ki-67 (%, median)	2	-	1	-	1	-	20	-

* Percentages are calculated using the respective subcategory (e.g., ‘Cystic’ or ‘Hypoenhancing’) as the denominator. Abbreviations: NET, neuroendocrine tumor; NEC, neuroendocrine carcinoma.

## Data Availability

Data supporting the results of this study were obtained from deidentified hospital records and are not publicly available due to privacy and confidentiality restrictions. Further inquiries can be directed to the corresponding author.

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
