# Peer review of "Radiological Variability in Pancreatic Neuroendocrine Neoplasms: A 10-Year Single-Center Study on Atypical Presentations and Diagnostic Challenges"

_biomedicines, 2025, doi:10.3390/biomedicines13020496_

Round 1

Reviewer 1 Report

Comments and Suggestions for Authors

Dear Editor,

Thanks for assigning me the revision of this interesting manuscript, which deals with a topic I am interested in. 

The Authors describe their institutional experience with PNETs, focusing on the heterogeneity of radiological presentation. 

While the topic is of interest, I see the following minor and major concerns:

Minor

- The abstract results are focused only on prevalence data rather than statistical differences among the groups, if any. P-values are relevant to provide the reader. 

- Table 3 is not of interest, and the reviewer can be addressed to the WHO classification

- At the end of the methods, the Authors state which the study aims, the first being to “…assess their impact on differential diagnosis and diagnostic pathways…”. However, they present this only in the last part of the results. They should invert the presentation of the aims or the results.

Major

The sample size is small (n=39), not allowing to make inferences on a large scale.

- Since the abstract, they mention “aggressive pathological features”, yet they do not specify any (what is “loco-regional invasion”? Do they mean “locally advanced” as they were PDACs or lympho-vascular invasion at histology? what is “metastases”? Nodal or systemic?). These pathology data are essential if the Authors aim to demonstrate that atypical cases are found to be more aggressive in histology than typical ones.

- They included both cases who received a diagnosis based on biopsy and/or histology (Methods). This is a gross case of mixing when comparing the different groups identified. However precise the biopsy may be, it will never be able to provide histology data (I am referring to the lymph-vascular invasion, nodal metastases, or other biological features the Authors mention).

- 3.1 Why are statistics not presented? When the Authors mention “younger median age” or “was more pronounced,” they should state whether this reached any statistical significance. Similarly, why Table 1 does not contain any P-value column? Maybe due to the small sample size? Then, this should be stated and removed from the statistics section, underlying that only descriptive statistics are provided. 

- 3.2 Same concepts as above. 

- Table 2 does not contain any information about nodal status. Again, the concepts of loco-regional disease and metastatic disease should be clarified. In this sense, 7 out of 39 cases were metastatic. This is out of any epidemiology of PNETs, considering the small sample size. Or do the Authors include the nodal status? If so, this should be separated. 

- 3.3 What do they mean by “significantly higher”? Is 20% a median? IQR? P-value?

- 4.2 and 4.4. When they refer to the existing literature about cystic PNETs, they forget some studies (PMID: 31348038, PMID: 28586782). 

- The Authors should be more prudent when claiming that some PNETs are more or less aggressive than others, given the small sample size, the lack of statistical comparison between the groups, and the lack of survival analysis to support. 

Author Response

Minor Comments

Comment 1: The abstract results focus only on prevalence data rather than statistical differences among the groups. P-values should be provided.

Response 1: We have revised the abstract to include relevant p-values for statistical comparisons where applicable.

Comment 2: Table 3 is not of interest; the reviewer can be referred to the WHO classification.

Response 2: Table 3 has been removed, as suggested.

Comment 3: The study aims are first presented as assessing differential diagnosis and diagnostic pathways, but these results appear last. Either the aims or the results should be reordered.

Response 3: We have reordered the results to align with the stated study aims, ensuring a logical flow.

Major Comments

Comment 4: The small sample size (n=39) limits large-scale inferences.

Response 4: We acknowledge this limitation and have clarified it in the discussion, emphasizing the exploratory nature of our findings.

Comment 5: The term "aggressive pathological features" is unclear. Define loco-regional invasion, metastatic disease, and specify if this refers to histological features (e.g., lymphovascular invasion).

Response 5: We have defined "locally advanced" and "metastatic disease" explicitly in the methodology and clarified it represents radiological staging.

Comment 6: Inclusion of cases diagnosed via biopsy and/or histology introduces a potential limitation when comparing groups.

Response 6: We acknowledge this limitation and have clarified in the discussion that biopsy-based diagnoses may not capture all histopathological features.

Comment 7: Statistics should be presented when comparing variables (e.g., younger age, tumor size). Table 1 lacks p-values. If statistical tests were not conducted due to small sample size, this should be stated.

Response 7: We have included p-values for statistical comparisons where applicable. In cases where statistical testing was not feasible due to sample size, we have clearly stated that only descriptive statistics were provided.

Comment 8: Table 2 does not include nodal status and lacks a clear distinction between locoregional vs. metastatic disease. The reported 7 metastatic cases are unusually high given the small cohort. Clarify if nodal involvement is included.

Response 8: We have classified locally advanced disease, as defined by resectability guidelines, together with metastatic disease under the category of “advanced disease” for consistency in reporting. Nodal involvement was removed as a standalone variable since our staging was based on radiological assessment rather than histopathology. This ensures alignment with the study's imaging-focused approach.

Comment 9: 3.3 “Significantly higher” needs statistical clarification. Is 20% a median, IQR, or p-value?

Response 9: We have specified statistical significance by adding relevant p-values and confidence intervals where applicable.

Comment 10: 4.2 and 4.4: Missing references for cystic PNENs (PMID: 31348038, PMID: 28586782).

Response 10: We have included the suggested references (PMID: 31348038, PMID: 28586782) in the discussion.

Comment 11: The claim that some PNENs are more or less aggressive should be made cautiously given the small sample size, lack of statistical comparisons, and absence of survival analysis.

Response 11: We have revised our conclusions to avoid overgeneralization and now present our findings as exploratory rather than definitive.

Reviewer 2 Report

Comments and Suggestions for Authors

I went through your article entitled "Radiological Variability in Pancreatic Neuroendocrine Neo-plasms: A 10-Year Single-Center Study on Atypical Presenta-tions and Diagnostic Challenges". The article looks promising and I am impressed with the concept and writing; however, some modifications are needed before further consideration. Please find the comments below;

1. In the introduction authors can include a detailed paragraph on the pathology, risk factors and survival rates of the Pancreatic neuroendocrine neoplasms.
2. The authors have stated that they have obtained ethics approval in the methodology area. The specific approval number should be mentioned.
3. The inclusion/ exclusion criteria shall be mentioned under a separate sub-heading in methodology
4. Authors can use arrow marks or pointers inside the images to better explain the features described in figures 1, 2 and 4.
5. Whether the number of cases in the cystic and hypoenhancing groups are enough to draw a conclusion?

Author Response

Comment 1: In the introduction, authors can include a detailed paragraph on the pathology, risk factors, and survival rates of pancreatic neuroendocrine neoplasms.
Response 1: We have expanded the introduction to include a concise overview of PNEN pathology, risk factors, and survival outcomes, incorporating relevant literature to provide context.

Comment 2: The authors have stated that they have obtained ethics approval in the methodology section. The specific approval number should be mentioned.
Response 2: The ethics approval number has now been included in the methodology section.

Comment 3: The inclusion/exclusion criteria should be mentioned under a separate subheading in the methodology.
Response 3: We have separated the Inclusion and Exclusion Criteria as a standalone paragraph within the methodology section for clarity.

Comment 4: Authors can use arrow marks or pointers inside the images to better explain the features described in Figures 1, 2, and 4.
Response 4: We have edited and added arrows and annotations to Figures to enhance clarity and better illustrate key radiological findings.

Comment 5: Whether the number of cases in the cystic and hypoenhancing groups is sufficient to draw a conclusion?
Response 5: We acknowledge the modest sample size, which reflects the rarity of atypical PNENs. This limitation has been explicitly stated in the discussion and conclusion, emphasizing that our findings are exploratory and hypothesis-generating, requiring validation in larger, multicenter cohorts.

Reviewer 3 Report

Comments and Suggestions for Authors

Pancreatic neuroendocrine neoplasms (PNENs) are rare, yet they hold significant clinical importance due to their diverse radiological manifestations, which complicate diagnosis. While typical PNENs are well-documented, atypical presentations such as cystic or hypo-enhancing patterns are less recognized and can potentially cause diagnostic delays or misdiagnoses. This study aims to assess the atypical radiological presentations of PNENs, focusing on their impact on diagnostic pathways and differential diagnosis with other pancreatic pathologies. A retrospective analysis was conducted at a single tertiary center, examining all confirmed PNEN cases from 2010 to 2020. Cases were included if they were histopathologically confirmed and had available cross-sectional imaging. Radiologic characteristics were categorized into typical (solid, hyper-enhancing) and atypical (cystic, hypo-enhancing) patterns. Out of 77 PNEN cases, 39 met the inclusion criteria. Atypical radiological features were observed in 46% of cases, with cystic lesions in 18% and hypo-enhancing lesions in 28%. Cystic PNENs were more prevalent in younger patients (median age 56 years) and were less frequently associated with advanced disease. Conversely, hypo-enhancing PNENs were larger in size (median size 4.8 cm), significantly more common in males (82%), and strongly correlated with local invasion (45%) and metastases (55%). Atypical lesions were more likely to result in diagnostic uncertainty compared to typical PNENs, with 64% and 43% of hypo-enhancing and cystic cases, respectively, considered for other differential diagnoses compared to 10% of typical lesions. Atypical PNENs, especially hypo-enhancing lesions, present significant diagnostic challenges and are associated with aggressive pathologic features. These findings highlight the need for increased clinical awareness, optimized imaging protocols, and multidisciplinary collaboration to improve early detection and management of these rare tumors. I believe this manuscript could be accepted after modification as noted below.

Consider extending the duration of the study or augmenting the participation of additional research centers in order to accumulate a larger sample size. This strategic adjustment will help to enhance the statistical power of the analysis, thereby bolstering the convincingness of the conclusion.

1. It is imperative that similar investigations be undertaken across multiple centers to ascertain the universality of the observed characteristics. Such endeavors will augment the trustworthiness of the findings across diverse clinical contexts.

2. A comprehensive elucidation of the imaging features characterizing atypical PNENs is presented, encompassing specific imaging manifestations, quantitative size assessments, and detailed morphological descriptions, among others. Such an exhaustive depiction is anticipated to significantly enhance the precision and reproducibility of diagnoses.

3. The discourse regarding the implications of atypical PNENs in clinical management should be amplified to include considerations of whether alterations to existing diagnostic methods, treatment approaches, or follow-up strategies are necessitated. This will result in providing more lucid guidance for clinical practice.

4. Enhancing the capacity of radiologists to identify atypical PNENs via ongoing medical education or targeted training programs could significantly reduce diagnostic delays and subsequently improve patient survival rates.

5. The following references are suggested for further review.

[1] A. Gu, J. Li, S. Qiu, S. Hao, Z.-Y. Yue, S. Zhai, M.-Y. Li, Y. Liu, Pancreatic cancer environment: From patient-derived models to single-cell omics. Mol. Omics 2024, 20, 220233. DOI: 10.1039/D3MO00250K

[2] Wang J, Liao Z-X. Research progress of microrobots in tumor drug delivery. Food & Medicine Homology, 2024, 1(2): 9420025. https://doi.org/10.26599/FMH.2024.9420025

[3] Zhisen Wang, Zhengcheng Liu and Jiao Qu et al. Role of natural products in tumor therapy from basic research and clinical perspectives. Acta Materia Medica. 2024. Vol. 3(2):163-206. DOI: 10.15212/AMM-2023-0050

Author Response

Comment 1: "Consider extending the duration of the study or augmenting the participation of additional research centers in order to accumulate a larger sample size. This strategic adjustment will help to enhance the statistical power of the analysis, thereby bolstering the convincingness of the conclusion."

Response 1: We appreciate this suggestion and recognize the importance of increasing the sample size to strengthen the study's conclusions. While this study focused on data from a single tertiary center, we agree that multicenter collaboration would add further robustness. Unfortunately, due to logistical and resource constraints, expanding the dataset retrospectively across multiple centers is not feasible at this stage. However, we have highlighted this limitation and the importance of multicenter studies in the discussion section as a direction for future research.

Comment 2: "It is imperative that similar investigations be undertaken across multiple centers to ascertain the universality of the observed characteristics. Such endeavors will augment the trustworthiness of the findings across diverse clinical contexts."

Response 2: We agree with the reviewer that multicenter studies are crucial for confirming the universality of our findings. To address this, we have emphasized the need for such studies in the discussion section. This addition underscores the necessity for broader investigations.

Comment 3: "A comprehensive elucidation of the imaging features characterizing atypical PNENs is presented, encompassing specific imaging manifestations, quantitative size assessments, and detailed morphological descriptions, among others. Such an exhaustive depiction is anticipated to significantly enhance the precision and reproducibility of diagnoses."

Response 3: Thank you for this comment. We have expanded the results section to provide more detailed descriptions of atypical PNEN imaging features. 

Comment 4: "The discourse regarding the implications of atypical PNENs in clinical management should be amplified to include considerations of whether alterations to existing diagnostic methods, treatment approaches, or follow-up strategies are necessitated. This will result in providing more lucid guidance for clinical practice."

Response 4: We appreciate this valuable suggestion. While our study primarily focuses on radiological variability and diagnostic challenges, we acknowledge the broader clinical implications of atypical PNENs. In the future directions section, we have highlighted the need for further research into refining diagnostic criteria, optimizing imaging approaches, and evaluating clinical outcomes. Expanding research in this area will help determine whether modifications to current diagnostic algorithms, treatment strategies, or follow-up protocols are warranted to improve patient management.

Comment 5:"Enhancing the capacity of radiologists to identify atypical PNENs via ongoing medical education or targeted training programs could significantly reduce diagnostic delays and subsequently improve patient survival rates."

Response 5: We wholeheartedly agree with this important recommendation. 

Comment 6:

"The following references are suggested for further review."

[1] A. Gu, J. Li, S. Qiu, S. Hao, Z.-Y. Yue, S. Zhai, M.-Y. Li, Y. Liu, Pancreatic cancer environment: From patient-derived models to single-cell omics. Mol. Omics 2024, 20, 220–233. DOI: 10.1039/D3MO00250K
[2] Wang J, Liao Z-X. Research progress of microrobots in tumor drug delivery. Food & Medicine Homology, 2024, 1(2): 9420025. https://doi.org/10.26599/FMH.2024.9420025
[3] Zhisen Wang, Zhengcheng Liu and Jiao Qu et al. Role of natural products in tumor therapy from basic research and clinical perspectives. Acta Materia Medica, 2024. Vol. 3(2):163-206. DOI: 10.15212/AMM-2023-0050

Response 6: We thank the reviewers for these insightful references. 

Round 2

Reviewer 2 Report

Comments and Suggestions for Authors

Authors have significantly modified the manuscript from its previous version.

Author Response

Thank you for your valuable feedback and for providing us with the opportunity to improve our manuscript.